# From Crafoord’s End-to-End Anastomosis Approach to Percutaneous Interventions: Coarctation of the Aorta Management Strategies and Reinterventions

**DOI:** 10.3390/jcm12237350

**Published:** 2023-11-27

**Authors:** Corina Maria Vasile, Gerald Laforest, Cristian Bulescu, Zakaria Jalal, Jean-Benoit Thambo, Xavier Iriart

**Affiliations:** 1Department of Pediatric and Adult Congenital Cardiology, University Hospital of Bordeaux, 33600 Bordeaux, France; gerald.laforest@chu-bordeaux.fr (G.L.); zakaria.jalal@chu-bordeaux.fr (Z.J.); jean-benoit.thambo@chu-bordeaux.fr (J.-B.T.); xavier.iriart@chu-bordeaux.fr (X.I.); 2Department of Medical and Surgical Cardiology for Congenital Heart Disease in the Fetus, Child, and Adult, Louis Pradel Hospital, 69677 Lyon, France; cristianbulescu@gmail.com; 3IHU Liryc—Electrophysiology and Heart Modelling Institute, Bordeaux University Foundation, 33600 Pessac, France

**Keywords:** coarctation of the aorta, recoarctation, surgical techniques, transcatheter intervention

## Abstract

First described in 1760 by the anatomist Morgagni, coarctation of the aorta (CoA) is a congenital heart defect characterized by narrowing the aorta, typically distal to the left subclavian artery. It accounts for approximately 5–8% of all congenital heart diseases, with an incidence estimated at 4 per 10,000 live births. In 1944, the Swedish surgeon Clarence Crafoord achieved the first successful surgical CoA repair by performing an aortic end-to-end anastomosis on two patients aged 12 and 27 years old. Presently, the most prevalent techniques for surgical repair, particularly in infants and neonates with isolated coarctation, involve resection with end-to-end anastomosis (EEA) and the modified Crafoord technique (extended resection with end-to-end anastomosis (EEEA)). Subclavian flap aortoplasty (SCAP) is an alternative surgical option for CoA repair in patients under two years of age. In cases where the stenosis extends beyond resection and end-to-end anastomosis feasibility, patch aortoplasty (PP) employing a prosthetic patch can augment the stenotic region, especially for older patients. Despite advances in pediatric cardiology and cardiac surgery, recoarctation remains a significant concern after surgical or interventional repair. This comprehensive review aims to provide a thorough analysis of coarctation management, covering the pioneering techniques introduced by Crafoord using end-to-end anastomosis and now extending to the contemporary era marked by percutaneous interventions as well as the recoarctation rate associated with each type.

## 1. Introduction

Coarctation of the aorta (CoA) presents ongoing challenges due to difficulties in prenatal diagnosis, anatomical variability, and long-term cardiovascular effects. Even after timely repair, patients remain at heightened cardiovascular risk. Regular hypertension screening and surveillance imaging are crucial in adults to monitor repair site complications [1].

First described in 1760 by the anatomist Morgagni, CoA is a congenital heart defect characterized by a narrowing of the aorta, typically occurring just distal to the left subclavian artery. It accounts for approximately 5–8% of all congenital heart diseases, with an incidence estimated at 4 per 10,000 live births [1,2,3]. Clinical presentation varies significantly depending on several factors, including the severity of the CoA and the presence of concomitant cardiac lesions, particularly those associated with left-sided heart obstruction [1]. In instances of severe CoA, neonates may manifest cardiovascular collapse, particularly upon ductal closure. Infants with CoA may present with a wide range of symptoms, from mild hypertension to severe heart failure, depending on the severity of the coarctation and associated cardiac anomalies [4].

In 1944, the Swedish surgeon Clarence Crafoord achieved the first successful surgical CoA repair by performing an aortic end-to-end anastomosis on two patients aged 12 and 27 years old [5,6]. Presently, the most prevalent techniques for surgical repair, particularly in infants and neonates with isolated coarctation, involve resection with end-to-end anastomosis (EEA) and the modified Crafoord technique (extended resection with end-to-end anastomosis (EEEA)) [7,8]. Subclavian flap aortoplasty (SCAP) is an alternative surgical option for CoA repair in patients under two years of age. In cases where the stenosis extends beyond resection and end-to-end anastomosis feasibility, patch aortoplasty (PP) employing a prosthetic patch can augment the stenotic region, especially for older patients [8].

Despite advances in pediatric cardiology and cardiac surgery, recoarctation remains a significant concern after surgical or interventional repair.

Our manuscript aims to provide a comprehensive review of the prognostic and risk factors for recoarctation after infantile CoA repair is rooted in the need for improved patient outcomes. Recoarctation can have serious consequences, including hypertension, left ventricular dysfunction, aortic aneurysm formation, and, therefore, the need for repeat surgery or catheterization. Identifying factors that predict or contribute to recoarctation is vital for early detection, intervention, and the development of personalized treatment strategies.

The primary objectives of this comprehensive review are to consolidate the existing data concerning infantile CoA repair and recoarctation, specifically focusing on identifying prognostic and risk factors. By systematically evaluating the current literature, this review aims to provide clinicians, researchers, and healthcare professionals in pediatric cardiology with a comprehensive understanding of the factors contributing to recoarctation. Ultimately, this knowledge is vital for improving clinical outcomes, optimizing patient care, and informing future research directions to pursue better outcomes and quality of life for individuals affected by this congenital cardiac anomaly.

## 2. Methodology

For this extensive review, we have conducted a comprehensive search of electronic databases, including PubMed, MEDLINE, Embase, and relevant medical journals, to identify peer-reviewed articles, systematic reviews, and meta-analyses related to infantile CoA repair and recoarctation. The search terms included “coarctation of the aorta”, “infantile coarctation repair”, “recoarctation”, and relevant variants.

Articles included in this review were selected based on their relevance to the topic, publication date (up to September 2023), and the availability of full-text articles. Studies focusing on pediatric populations and those reporting on prognostic and risk factors for recoarctation were considered. Non-English language articles were excluded.

Data extraction was performed systematically, including information on study design, patient demographics, surgical techniques, follow-up protocols, outcomes, and key findings related to prognostic and risk factors.

The quality of each selected study was assessed using established criteria for observational studies, randomized controlled trials, and systematic reviews. This assessment included study design, sample size, methodological rigor, and potential biases.

## 3. Coarctation Management

### 3.1. Surgical Approach

#### 3.1.1. Resection with End-to-End Anastomosis

In 1945, Crafoord and Gross described the first successful surgical repair through a left lateral thoracotomy [5,9]. This procedure involved the adequate mobilization of the descending aorta, isthmus, and distal arch, followed by the clamping of the aorta proximal and distal to the coarctation. The arterial ligament (or the ductus arteriosus, if it is permeable) was also ligated and transected close to the aorta. The coarcted segment was then resected. Subsequently, the remaining aortic arch just distal to the left subclavian artery and descending aorta were connected with a direct end-to-end anastomosis. However, this initial repair was associated with a relatively high incidence of recoarctation, with reported rates ranging from 41% to 51% [10,11,12,13]. Recoarctation was found to be age-dependent and most pronounced when the surgery was performed in neonates. Given the high incidence of recoarctation, direct end-to-end anastomosis is not commonly utilized in contemporary surgical practice.

#### 3.1.2. Subclavian Flap Repair

This technique was initially described by Waldhausen and colleagues in 1966 [14]. The subclavian artery is dissected and ligated near the origin of the left vertebral artery. Creating a flap involves making an incision on the lateral wall of the subclavian artery, which is extended downwards over the aortic isthmus and across the stenosed segment. This flap is then folded downward and securely sutured into the incised aorta, enlarging the narrowed aortic region. When this technique is performed in older children, the rate of recoarctation appears to be relatively low, ranging from 0% to 3% [15,16]. However, when applied to neonates, recoarctation may be as high as 23% [17]. It is worth noting that although sacrificing the subclavian artery does not typically result in left arm ischemia, it may lead to claudication in the long term [18].

#### 3.1.3. Interposition Graft

Introduced by Gross in 1951 [19], this approach involves excising the coarctation segment and subsequently employing a Dacron tube graft or aortic homograft interposition. Given the utilization of non-native tissue, this technique cannot grow alongside the patient, making it primarily suitable for cases where graft outgrowth is not a concern. In the context of adults, both short- and long-term outcomes are commendable: Yousif and colleagues documented a peri-operative mortality rate of 0% and zero instances of recoarctation over a mean follow-up period of 10 ± 7.6 years [20]. Similar favorable results have been reported by various other research groups [21].

#### 3.1.4. Patch Angioplasty

Given the elevated risk of recoarctation associated with the end-to-end anastomosis technique, medical professionals began exploring alternative surgical approaches for addressing aortic coarctation. Vosschulte was a pioneering surgeon who initially introduced the concept of utilizing a prosthetic patch to enhance the aorta [22]. This procedure involves making a longitudinal incision in the affected segment and then suturing a prosthetic patch across the incision to reinforce and expand this area. Although the use of Dacron grafts resulted in a decreased rate of recoarctation [23], it also presented a notable drawback—a high incidence of aortic aneurysm formation, ranging from 20% to 40%. Subsequently, adopting polytetrafluoroethylene materials reduced the occurrence of aneurysms to 7%, albeit at the cost of a higher recoarctation rate, which reached 25%. Consequently, patch aortoplasty has largely fallen out of favor for treating uncomplicated aortic coarctation [24]. Nevertheless, it still finds application in complex cases necessitating aortic arch reconstruction.

#### 3.1.5. Extended End-to-End Anastomosis

Amato introduced this technique in 1977, and it remains a commonly employed procedure in contemporary practice [25]. In contrast, to direct end-to-end anastomosis, this method involves clamping the proximal aorta across the aortic arch, which may include the left subclavian or even the left carotid artery along with the aortic arch. Distally, the aorta is clamped below the stenosed segment. After ligating and dividing the ductus arteriosus, the coarctation segment is excised, and the inferior aspect of the aortic arch is opened. A counter-incision is performed on the lateral wall of the descending aorta. The subsequent end-to-end anastomosis is larger than in the classic technique described by Crafoord, and it allows simultaneous enlargement of a moderately hypoplastic aortic arch. This procedure can be conducted with low peri-operative mortality rates, and reports indicate relatively low recoarctation rates ranging from 4% to 13% [26,27,28,29,30,31,32].

#### 3.1.6. Single vs. Stage-Management Approach for Coarctation of the Aorta with Associated Congenital Cardiac Anomalies

When coarctation of the aorta is diagnosed in conjunction with other congenital cardiac anomalies, clinicians must carefully consider the optimal treatment strategy. Most patients we encounter in this setting are newborns presenting with coarctation and either a ventricular septal defect (VSD), transposition of the great arteries, truncus arteriosus, or atrioventricular septal defect (AVSD) [33]. The single-stage approach involves repairing all cardiac defects at once, including the coarctation. This method offers the advantage of addressing all anomalies in a single procedure, potentially minimizing the need for future interventions. Additionally, this strategy allows for extended arch repair under hypothermic circulatory arrest and antegrade cerebral perfusion in cases where a hypoplastic arch is also present.

However, the single-stage approach can be technically challenging and may carry a higher risk of complications, such as prolonged cardiopulmonary bypass time or excessive bleeding.

Alternatively, the stage-management approach involves addressing the coarctation separately from the other cardiac anomalies. This allows for a more focused and controlled repair of the coarctation, thereby reducing the complexity of the procedure. By staging the interventions, clinicians can carefully monitor the patient’s response to each procedure and assess the need for further interventions based on individualized factors such as age, anatomy, and hemodynamics. For example, the most common association is the coarctation of the aorta with VSD. The likelihood of spontaneous VSD closure is a main determinant in choosing a therapeutic protocol. Some studies have shown that small and muscular VSDs are the most likely to close spontaneously, as opposed to larger defects or those located elsewhere (such as perimembranous, outlet, or malalignment defects) [29]. Practically speaking, if the defect is deemed unlikely to close on its own, the initial intervention should also address this lesion.

The two-stage repair approach involves performing coarctation repair and pulmonary artery banding through a left thoracotomy, followed by VSD closure at a later date. This approach is advantageous in cases without proximal arch hypoplasia, as it allows for a straightforward coarctation repair and a VSD closure at a later moment, when intracardiac repairs carry fewer risks of complications, such as AV block. However, some patients may experience spontaneous VSD closure, potentially obviating the need for a second operation. Disadvantages of this approach include a period of palliation between procedures and potential complications associated with the pulmonary band. Conversely, the single-stage approach involves performing coarctation repair and VSD closure simultaneously on cardiopulmonary bypass, with either circulatory arrest or regional perfusion during coarctation repair. This method offers the benefits of complete repair in infancy without the need for palliation and the ability to address proximal arch hypoplasia. Nonetheless, it is a technically more challenging operation and requires either circulatory arrest or regional cerebral perfusion. Some studies have suggested a higher risk of recoarctation with this approach. An alternative method is coarctation repair without cardiopulmonary bypass through a thoracotomy, followed by VSD closure during the same operation (one stage, two incisions). This approach provides excellent clinical outcomes and allows for complete repair in infancy. It also avoids prolonged periods of aortic cross-clamping, cardiopulmonary bypass, and circulatory arrest/regional perfusion. Compared to other strategies, this approach is associated with shorter total intensive care units and hospital stays [34].

In conclusion, the choice between the single-stage and staged-management approaches should be made on a case-by-case basis, considering the patient’s specific characteristics and associated cardiac anomalies.

### 3.2. Transcathether Interventions

#### 3.2.1. Balloon Angioplasty

Transcatheter balloon angioplasty for native CoA made its debut in the early 1980s [35]. This procedure involves the placement of a balloon catheter across the coarctation site, typically via a retrograde approach, although an antegrade approach is also utilized. During the procedure, precise angiographic measurements are conducted to assess the dimensions of the coarctation site and the aorta proximal and distal to the lesion. Based on these measurements, an appropriately sized balloon is selected to dilate the narrowed area [36]. The primary objective of this intervention is to induce a controlled tear in the intima and media layers by carefully stretching the constricted vessel segment. Subsequently, remodeling the aortic wall is anticipated to lead to a sustained resolution of CoA and prevent the risk of recoil [37].

Balloon angioplasty is often the preferred option for older children [16]. It is also the treatment of choice for younger patients who experience recoarctation. However, it is important to note that the use of balloon angioplasty in neonates and very young infants is typically reserved for cases where associated ventricular dysfunction is present, to stabilize the patients for subsequent definitive surgical repair [38]. The utilization of balloon angioplasty as the primary intervention in this extremely young age group has become less common due to its associated high recurrence rate and the potential risk of vascular complications [38].

In specific cases where neonates are too clinically compromised to undergo immediate surgical intervention, balloon angioplasty emerges as a palliative measure to stabilize their condition [39,40]. Nonetheless, the percutaneous treatment of aortic coarctation in neonates and infants remains contentious due to concerns surrounding residual or recurrent stenosis and the potential formation of aneurysms at the dilation site [41]. Urgent balloon dilation, however, has demonstrated its ability to reduce mortality rates significantly, serving as a bridge to surgical intervention for severely ill patients [42,43].

The procedure typically involves femoral artery access. Nevertheless, alternative access routes, such as carotid or axillary artery access, may be considered in cases involving low-body-weight infants. These alternative routes offer a direct, antegrade trajectory to the aortic isthmus, presenting several technical advantages. Notably, the axillary artery route is advantageous in smaller patients, including premature neonates and those with critical aortic coarctation, where femoral pulses may not be palpable. Additionally, axillary access is crucial in concomitant low cardiac output cases, where locating a femoral pulse can be challenging. Importantly, unlike carotid artery access, the axillary route is not an end artery and does not necessitate surgical cutdown and repair [43].

Balloon sizing is based on choosing an initial balloon diameter two to three times the minimal CoA diameter, ensuring it does not exceed the diameter of the aorta at the diaphragm. Typically, the procedure involves two or three dilations, each with a brief inflation time lasting less than 10 s.

Balloon angioplasty has shown remarkable acute success in infants under three months of age, with an approximate 50% restenosis rate, particularly in cases without aortic arc hypoplasia and lower reintervention rates [44]. A recent retrospective study involving 68 patients with native aortic coarctation conducted by Sandoval et al. [45] demonstrated the effectiveness and safety of balloon angioplasty in infants aged 3 to 12 months. Their findings indicated outcomes comparable to those observed in older children and adults. As a result, repeat angioplasty or stent placement can often obviate the need for surgical intervention.

#### 3.2.2. Stent Implantation

Transcatheter stent implantation made its debut in the late 1980s and gained widespread acceptance as a therapeutic approach for CoA patients in the early 1990s [46]. This method is the preferred treatment for both native and recurrent CoA in older children, adolescents, and adults [47,48]. However, it presents technical intricacies compared to balloon angioplasty, necessitating larger vascular sheaths for access [49]. Once securely positioned within the aorta, the stent evenly distributes radial force and provides sustained relief from the gradient [50,51,52]. Findings from the Congenital Cardiovascular Interventional Study Consortium (CCISC) and the Coarctation of Aorta Stent Trial (COAST) trials reveal that patients who undergo stent implantation experience a lower rate of acute complications compared to their counterparts who undergo surgery or balloon angioplasty [50,51]. Nonetheless, they are more likely to require planned reinterventions for stent dilatation, particularly when the procedure is performed on younger patients [15]. Acute complications stemming from stent implantation encompass stent migration, stent embolization, vessel “jailing,” and aortic dissection [15,53]. Long-term complications entail planned reinterventions for stent dilatation, neo-intimal proliferation within the stent leading to stenosis, stent fracture, and the formation of aneurysms [15,53]. The utilization of covered stents has demonstrated a decreased incidence of aneurysms and dissection following stent implantation [54].

A study conducted by Forbes et al. between 2002 and 2009, encompassing data from 36 institutions, compared the safety and efficacy of surgical and transcatheter treatment options for native coarctation, both in the acute phase and during follow-up [53]. Their observations revealed that both surgical and stent therapies achieved lower upper-lower extremity blood pressure gradients compared to balloon angioplasty, both acutely and in the short term. However, these differences began to diminish in the intermediate follow-up period. Notably, patients undergoing stent implantation experienced fewer acute complications when juxtaposed with those undergoing the other two treatment modalities. However, a higher proportion of stent patients required planned interventions in the future [15]. It is crucial to acknowledge that a younger age at the time of intervention correlates with a heightened risk of CoA recurrence [55]. Nonetheless, a similar survival rate of approximately 93% at 10 years, 86% at 20 years, and 74% at 30 years postoperatively has been reported in older and contemporary postsurgical cohorts [54,55]. Despite advances in management techniques and medical care, the long-term survival rates have mostly stayed the same.

In neonates and young infants with CoA, stent implantation is considered an alternative intervention to address the condition. This approach is typically favored when a percutaneous procedure is deemed more suitable. Stent implantation using pre-mounted coronary stents has gained prominence as a safe option to alleviate acute symptoms and serve as a bridge to surgery [56,57], particularly since it avoids the need for larger sheaths associated with stent delivery that poses challenges in very young patients.

Stent implantation offers advantages in maintaining vessel patency and can be a viable option for preterm neonates with CoA. Typically, surgical repair, including stent removal, is undertaken once the patient reaches an adequate body weight and overall condition for the procedure. However, it is essential to acknowledge that stents can cause significant tissue trauma and loss during surgical removal [58].

Recent advancements have introduced bioresorbable stents as a promising alternative with the potential for reduced tissue damage during surgical extraction. Nevertheless, their applicability in very-low-weight patients with aortic coarctation requiring long-term bridging to surgery remains under scrutiny, as early experiences have noted restenosis and stent failure attributed to the radial force loss in the scaffold [59]. Further research is warranted to assess bioresorbable stents’ feasibility and long-term outcomes in this patient population.

To sum up, surgical repair, particularly the extended end-to-end anastomosis technique, is most of the time the preferred surgical method for treating coarctation of the aorta. This approach is favored because it avoids using prosthetic material, allowing for the resection of the coarctation segment. Moreover, the extended end-to-end anastomosis involves a wider incision, which is less prone to restenosis. This surgical method is typically the preferred choice, especially in cases of native coarctation in infants and young children, patients requiring repair of associated cardiac defects, or individuals with complex coarctation anatomy.

In cases where coarctation reoccurs after the initial repair, balloon angioplasty often becomes the preferred intervention. It is a minimally invasive procedure involving inflating a balloon in the narrowed segment of the aorta to widen it. However, there is a concern for recoarctation and the formation of aneurysms in patients with native coarctation who undergo balloon angioplasty. The long-term effectiveness of this approach can vary, and close follow-up is crucial to monitor for potential complications.

Endovascular stent placement is another interventional option that provides structural support to the narrowed aortic segment. In some cases, it is preferred over balloon angioplasty because it is associated with decreased rates of aortic wall injury and aneurysm formation. Covered stents with a fabric covering may be used to protect against shear stress and prevent subsequent restenosis. However, it is essential to exercise caution when using stents to avoid covering vital branch vessels, as this can lead to complications.

Using stents in small children is a topic of debate, as it often requires larger sheath sizes, which can be challenging in young patients. Additionally, the limitations in accommodating somatic growth need to be carefully considered when choosing this approach, as children’s bodies grow and change over time, potentially affecting the stent’s effectiveness and fit. Each intervention method has its advantages and limitations, and the choice of approach should be based on the patient’s specific condition and individual factors. Regular follow-up is essential to ensure the chosen treatment’s success and address potential issues that may arise over time [60].

In general, the approach to coarctation management hinges on factors such as the patient’s age at presentation, the complexity of the coarctation, and whether it is a native or recurrent obstruction [61,62,63,64,65] (Figure 1).

For infants and young children with native coarctation, surgical repair is often preferred due to the reduced long-term risk of aneurysms compared to balloon angioplasty, the need for subsequent stent re-dilations, and the limitations posed by small arteries that cannot accommodate larger sheath sizes [61,64]. However, balloon angioplasty can be a palliative strategy in emergencies involving extremely ill neonates when immediate surgical intervention is not feasible [64]. Surgical repair is also suitable for cases with complex coarctation anatomy or when addressing associated cardiac defects is necessary.

In contrast, for older children, adolescents, or adults with a simple, juxtaductal, native coarctation, stent placement is a reasonable and less invasive alternative to surgery, with favorable long-term outcomes [61,64,65]. Using stents that can expand to an adult size is important to avoid needing later surgical intervention.

Initial balloon angioplasty is a reasonable option for recurrent coarctation in younger children since the long-term risk of aneurysms is lower than with native coarctation [64]. However, the success of balloon angioplasty varies, and surgical reintervention may be necessary if the obstruction relief is incomplete [62]. Stent placement is also a consideration for recoarctation in older children and adolescents, particularly when the stent can be dilated close to adult size, minimizing the need for multiple re-dilations [65].

## 4. Comparative Analysis of Aortic Coarctation Repair Techniques

In a 1994 study conducted by Kapetein et al. [66] on 109 patients operated between 1953 and 1985, with 17% having isolated coarctation and a mean age of 11 ± 12 months at the time of surgery, aortic coarctation resection was performed due to nonelective conditions such as congestive heart failure or severe systemic hypertension. Of these patients, 48 underwent classic end-to-end anastomosis with silk sutures, whereas 26 had extended end-to-end anastomosis with polypropylene. The post-operation recoarctation rate was relatively low at 5.8%. However, long-term follow-up revealed that recoarctation occurred in 30 (41%) discharged patients. Over a 30-year follow-up period, Kaplan-Meier estimates showed an 86% rate of recoarctation in patients who had classic end-to-end anastomosis with silk sutures (n = 48), whereas none in the group with an “extended” anastomosis and polypropylene sutures (n = 26) experienced recoarctation. Cox analysis indicated that age at operation below 6 months was a prognostic factor for recoarctation. The extended anastomosis with polypropylene sutures was not a significant prognostic factor for recoarctation due to the shorter follow-up duration.

In a 2010 retrospective study by Dehaki et al. [67], involving 188 patients below the age of 14 years who underwent surgical repair for CoA between 1994 and 2004, with an average follow-up of 81.6 ± 32.8 months and a mean age of 5.4 ± 4.2 years, the recurrence rate was 10%. The median time for recurrence was 3.5 years after the primary repair. The repair methods in this study included patch repair in 59% of cases (using either Goretex or Dacron), end-to-end anastomosis in 20.7%, and subclavian flap repair in 16.5%. Notably, angiographically documented recurrence was more common in patch repair (12.7%) and end-to-end anastomosis (10.3%) compared to subclavian flap repair (3.2%).

According to a 2010 Turkish study led by Uguz et al. [68], involving 91 patients (35 neonates and 56 infants) with an average follow-up of 44 months, recoarctation occurred in 13 patients (12.1%). Within this group, eight neonates who had undergone primary operations experienced a 22.9% recoarctation rate. In contrast, five patients operated on during infancy (within the first 6 months of life) exhibited an 8.9% restenosis rate. Recoarctation developed within the first year after the initial repair in all 13 cases. Out of these 13 patients, 10 (11% of the entire study population) required reintervention between six months and two years after the initial repair. Seven (including four neonates) underwent successful balloon dilatation for their recoarctation, whereas three (including two neonates) had successful reoperations. None of these 10 patients displayed signs of stenosis recurrence during their last follow-up. In the remaining three patients (two neonates), only mild recoarctation emerged, with 22, 24, and 27 mmHg gradients, respectively, which did not necessitate treatment. Importantly, no children who were operated on between 6 and 12 months of age exhibited recoarctation throughout the follow-up. Out of the 91 patients, only 7.7% required interventions for recoarctation, with 3.3% undergoing surgery. Regarding the surgical technique used for coarctation repair, 7.1% of patients who underwent extended end-to-end anastomosis experienced recoarctation. In comparison, only 2.9% of patients with end-to-end anastomosis repairs encountered recoarctation.

In a retrospective study by Jahangiri and colleagues [69], the early and long-term outcomes of subclavian flap angioplasty in neonates and infants were assessed. The study included 185 consecutive patients who underwent this procedure between 1974 and 1998, comprising 125 neonates and 60 infants, with a median age of 18 days. Among the patients, 66 (36%) had an additional ventricular septal defect, 41 (22%) were diagnosed with aortic arch hypoplasia preoperatively, 141 (76%) had an associated patent ductus arteriosus, and 41 (22%) had additional complex heart diseases. Follow-up assessments were performed using transthoracic Doppler echocardiography. The study reported an early mortality rate of 3% and identified recoarctation, defined as a Doppler gradient of 25 mm Hg or more, in 11 (6%) patients during a median follow-up period of 6.2 years (6.2 ± 4.6 years). Importantly, no complications related to the left arm were noted. Risk factors for mortality were found to be residual arch hypoplasia and low birth weight. In contrast, the persistence of arch hypoplasia after surgical treatment was the sole risk factor for recoarctation. However, it was determined that recoarctation was likely not due to a hypoplastic transverse arch but rather at the site of ductal tissue. The study showed excellent survival rates, with 98% and 96% survival at 5 and 10 years, respectively. The freedom from reoperation for recoarctation was notably high, with rates of 95% at 2 years and 92% at 5, 10, and 15 years. The study reaffirms that subclavian flap repair remains an effective and low-mortality technique for aortic coarctation repair. Furthermore, it highlights that arch hypoplasia tends to regress in most patients after this procedure, underlining the effectiveness and long-term benefits of this surgical approach.

Another noteworthy study was conducted by Burch PT and colleagues [70] in the USA, involving a cohort of 167 neonates and infants aged less than 90 days, with a mean age of 16 days and a median weight of 3.4 kg. All patients underwent end-to-end anastomosis repair using various running sutures, including 6-0 or 7-0 polydioxanone sutures or 7-0 or 8-0 polypropylene sutures. Among the 125 patients who had the anastomosis performed with polydioxanone sutures, 9 (7.2%) experienced recurrent or residual coarctation, whereas 8 (19.5%) of the 41 individuals who underwent repair with polypropylene sutures had recurrence (*p* = 0.04). This study emphasized that low weight does not affect survival or reintervention rates after coarctation repair in neonates and infants less than 3 months of age.

Data from an Indian study, conducted by Sen S. et al. [71] in 2018 on 75 patients (34 neonates) who underwent coarctation repair. A total of 28 patients underwent balloon angioplasty, and 47 patients underwent surgical repair (23 patients had pericardial patch coarctoplasty, 21 underwent resection and end-to-end anastomosis, 2 patients went for subclavian flap coarctoplasty, and only 1 patient for interposition graft). For a more in-depth analysis of risk factors for reintervention, patients were categorized into neonates (age 0–1 month) and infants (age > 1–12 months) at the time of presentation. Among neonates, 63.6% underwent balloon angioplasty, and 17.4% required surgical correction. Concerning the occurrence of recoarctation, patients who underwent balloon coarctoplasty were associated with a significantly higher rate of reintervention (Chi-squared test *p* = 0.007). In infants aged 1 month to 1 year, there was no significant difference in the choice of surgical technique and the rate of recoarctation.

In 1995, Rao P.S. and colleagues [72] performed a study on 29 infants diagnosed with native coarctation, no older than 3 months, in which they compared the outcomes based on the intervention type. A total of 14 patients underwent surgical correction and 15 had balloon angioplasty. The two groups had no significant difference based on the patients’ mean weight. Recoarctation occurred in 46% of patients [6] from the group that underwent surgical correction during a mean follow-up of 4.5 years. For reintervention, all patients underwent balloon angioplasty. Regarding the patients that underwent balloon angioplasty as a primary intervention, recoarctation occurred in 50% (7 children) during a mean follow-up of 4 years. Two underwent surgical correction, and the other five underwent repeat balloon angioplasty.

Another interesting study by Mohan et al. [73] has provided valuable insights into using stents for CoA treatment in children weighing less than 30 kg. Their research included 60 children divided into two groups: 22 patients weighing less than 30 kg in the first group and 38 patients weighing 30 kg or more in the second group. The study’s findings were notable. After stent implantation, there was a significant increase in the mean minimum diameters of the CoA and the CoA/descending aorta (CoA/DAo) ratio. Simultaneously, there was a considerable reduction in the mean systolic gradient in both groups post-stent intervention. Importantly, no significant differences were observed between the two groups in key metrics, including the CoA/DAo ratio, residual systolic gradients, and the reduction in systolic gradient following stent implantation. Furthermore, the study reported no significant complications in patients weighing less than or equal to 30 kg. In summary, Mohan et al.’s [73] research emphasizes the effectiveness and safety of using large stents for short-term CoA management in small children. Although the short-term outcomes are promising, it is crucial to note that these patients may require stent redilations, necessitating ongoing and vigilant follow-up. This study significantly enhances our understanding of CoA treatment strategies in this specific patient population, providing valuable insights for clinicians and affirming the feasibility and safety of using large stents in managing CoA in small children.

Corno et al. [74] conducted a study over 30 years involving 141 patients, including neonates, infants, children, and adults. The most common CoA repair methods among adults included patch aortoplasty, resection, and end-to-end anastomosis, pyloroplasty type, and other procedures. There was no recoarctation observed in the adult group. In the children’s group, resection and end-to-end anastomosis were most frequently performed, followed by patch aortoplasty, subclavian flap repair, and other techniques. The recurrence rates varied depending on the approach, with end-to-end anastomosis at 1.8%, patch aortoplasty at 25.0%, and subclavian flap repair at 21.4%. Ten patients required surgical reoperation.

Padalino et al. [75] reported on a study with a median follow-up of 10.2 years, involving 341 patients with a median age of 25 days. The most common CoA repair approach was extended end-to-end anastomosis in 80.9% of cases, standard end-to-end anastomosis in 14.7%, and other techniques in 4.4%. The recoarctation rate was 4.5%, and percutaneous procedures such as stenting or balloon dilatation were used for management.

Kaushal et al. [76] conducted a study with a mean follow-up duration of approximately 5.0 years, involving 201 patients, including neonates with a median age of 23 days. Their study’s primary CoA repair approach was extended end-to-end anastomosis in 4% of cases. Three patients underwent balloon angioplasty, and five required reoperation.

Presbitero et al. [77] conducted an extensive long-term follow-up study, spanning an average duration of 20 years, involving 244 patients. Among this cohort, 143 patients were subject to follow-up assessments. The primary methods for aortic coarctation repair encompassed various surgical techniques, including end-to-end anastomosis, extended end-to-end anastomosis, prosthetic tube graft, and patch repair. Within this cohort, 15 patients experienced CoA recurrence, necessitating recoarctation management. The study by Presbitero et al. [77] focused on assessing the long-term outcomes in 226 patients who had undergone surgical repair for aortic coarctation. The evaluation was conducted 15 to 30 years after the surgical intervention, during which 26 patients did not survive the follow-up period, primarily due to causes related to the surgical repair itself or associated cardiovascular anomalies. The study’s findings revealed that the survival rates for patients who had surgery between the ages of four and 20 years were 97%, 97%, and 92% at 10, 20, and 30 years post-operation, respectively. In contrast, for patients operated on after the age of 20, the corresponding survival rates were 93%, 85%, and 68%. Importantly, a statistically significant difference in survival rates became evident from the fifteenth year of follow-up. Notably, the survival of patients operated on before age 20 did not significantly differ from that of a comparable general Italian population. Regarding CoA management outcomes, recoarctation occurred in only 8% of patients who underwent end-to-end anastomosis, whereas it was observed in 35% of those who had undergone other surgical procedures. A significant observation was that two-thirds of the patients were hypertensive at the last visit. The actuarial curve demonstrated that although blood pressure was normal in most patients 5 to 10 years after the operation, the long-term prognosis indicated that only 32% of patients are expected to have normal blood pressure 30 years after coarctation repair. In conclusion, the study suggests that early surgical repair for aortic coarctation significantly enhances long-term survival. The research also highlights that intervention in older patients and in high blood pressure cases are pivotal predictors of late-onset hypertension. These findings underscore the importance of timely and effective surgical repair for aortic coarctation, emphasizing its potential impact on long-term outcomes [78].

In a retrospective study by Josep Rodes-Cabau et al. [78], 80 patients aged over 1 year with isolated CoA were analyzed for outcomes following either surgical repair (with 77% having an end-to-end anastomosis, 17% receiving a patch aortoplasty, and 9% undergoing an interposition graft) or transcatheter angioplasty. The mean follow-up was approximately 3 years, during which 30 patients received surgical repair and the rest underwent angioplasty. Immediate hemodynamic results were similar, but surgical patients experienced higher complication rates (50% vs. 18%) and longer hospitalization (7 days vs. 1 day). Notably, aortic aneurysms were more common in the angioplasty group (24% vs. 0%), with some requiring further intervention. Although no surgical patients needed recoarctation repair, 32% of the angioplasty group required reintervention, but both groups showed similar blood pressure control at the latest follow-up.

Among the 30 patients who underwent surgical repair, the procedures varied, with 77% having an end-to-end anastomosis, 17% receiving a patch aortoplasty, and 9% undergoing an interposition graft. In contrast, among the 50 patients who received transcatheter treatment, 62% underwent balloon angioplasty, and 38% had stent implantation. Various types of stents were utilized in the stent implantation cases, including Palmaz 308, Palmaz 4014, Genesis 2910, Cheatham-Platinum, and covered Cheatham-Platinum. Additionally, five patients in the angioplasty group required staged dilation procedures to achieve optimal results.

For a detailed understanding of recent research, Table 1 summarizes the key findings from studies on CoA.

## 5. Long-Term Outcomes after Coarctation Repair

After undergoing surgical or percutaneous repair, assessing the long-term outcomes for patients is crucial.

Even after successful enlargement of the aorta to an adequate size, patients who have undergone treatment for coarctation face a higher risk of cardiovascular events (such as atrial arrhythmia, heart failure, ventricular tachycardia, and stroke) and mortality compared to the general population [79,80,81,82,83,84,85,86]. A major risk factor in these patients is the persistence of hypertension post-repair, which occurs in 44% to 60% of adults who have had aortic coarctation repaired [87].

The shape of the aortic arch is a significant factor in this context. Variations in the arch’s architecture, described as gothic, crenelated, or normal, along with a reduced size of the aorta at the site where the stent is placed, are critical in developing hypertension post-repair [88,89]. These factors contribute significantly to the increased cardiovascular risks faced by these patients [88].

The unique characteristics of the aortic arch can significantly impact hemodynamic forces and blood flow distribution, leading to blood pressure variations. Specifically, the recurrence or continued presence of CoA can trigger or exacerbate systemic hypertension and its related complications [89].

This correlation emphasizes the necessity for lifelong monitoring and comprehensive evaluation of patients with aortic coarctation. The key to this process is CT or MRI imaging for anatomical assessment and vigilant hypertension management. This involves continuous 24 h blood pressure monitoring in the right arm or using Ambulatory Blood Pressure and Exercise Testing. The latter includes measuring peak systolic blood pressure during exercise, independently associated with cardiovascular events [86,89,90]. This approach is vital in the follow-up of patients with aortic coarctation, as it offers critical predictive insights into cardiovascular risk. These approaches align with European and American guidelines [88,91].

Such monitoring is essential for identifying patients requiring antihypertensive treatment even without recoarctation [89,90]. Persistent hypertension in these patients is attributed to a combination of factors: endothelial dysfunction, abnormal reactivity of the arterial smooth muscle, changes in the material properties of the aorta leading to increased arterial stiffness, and decreased baroreceptor sensitivity, which results in heightened sympathetic activation [80,81,86].

Consequently, chronic hypertension imposes an overload on the left ventricle, causing increased stiffness, myocardial fibrosis, and dysfunction in both systolic and diastolic phases. These pathological changes manifest clinically as atrial fibrillation and heart failure, stemming from diastolic dysfunction and remodeling of the left atrium. Additionally, ventricular tachycardia can occur due to scarring and fibrosis of the left ventricular myocardium [81,89]. These developments underline the critical importance of evaluating and managing blood pressure response to exercise in patients with a history of aortic coarctation.

It is crucial to understand how aortic stiffness can influence central arterial pressure. Each time the left ventricle ejects blood, an incident wave is transmitted through the conduction arteries and then reflected at several impedance mismatch locations in the arterial system. The arrival time of the reflected waves at the proximal aorta depends on the aortic stiffness of the conduction arteries, which transmit both the forward and reverse displacement waves. In a healthy aorta, the pulse wave velocity is slow, and the reflected wave arrives at the proximal aorta during diastole, producing an increase in diastolic central arterial pressure without a significant increase in systolic central arterial pressure [81].

On the other hand, in a rigid aorta, the reflected waves reach the proximal aorta at the end of systole, producing an increase in systolic central arterial pressure (left ventricular afterload) and a loss of increase in diastolic central arterial pressure (coronary perfusion pressure) [81].

The pathophysiology of this late-onset hypertension is at least partially linked to dysfunction of the conduction arteries of the upper limbs. Previous studies have demonstrated abnormalities in elastic properties and reactivity in the conduction arteries of the upper limbs, which are absent in the arteries of the lower limbs. The decrease in glyceryl trinitrate-dependent dilation (GTND) may be attributed to abnormal relaxation of smooth muscle cells or structural abnormalities of the arterial wall, such as increased collagen and reduced elastin [83,92].

It is important to understand that even after a successful repair of aortic coarctation, without any residual obstruction or aortic aneurysm, the vascular reactivity and mechanical properties of the large conduction arteries remain impaired [83]. This ongoing issue can lead to long-term vascular health problems in these patients, highlighting the importance of continued monitoring and early intervention to reduce potential complications.

Studies have shown that the timing of the repair surgery affects various aspects of vascular function differently [83]. Patients who had aortic coarctation repair within the first four months of life showed normal arterial stiffness (measured by pulse wave velocity) but had impaired brachial artery reactivity, as indicated by reduced responses in brachial FMD (Flow-Mediated Dilation) and NTG (Nitroglycerin) tests [83].

Early repair seems to maintain the elasticity of the conduction arteries. This observation partly explains the established link between the timing of the repair and the patient’s prognosis. However, despite early repair, functional abnormalities at the endothelium level and smooth muscle cells remain a concern. This justifies the need for careful, long-term monitoring of patients undergoing aortic coarctation repair [83].

The persistent abnormalities in the upper limb’s conduction arteries, observed during long-term follow-up, indicate that aortic coarctation is linked to widespread arterial dysfunction. It seems that these vascular changes are partly due to the abnormal hemodynamics in the upper body before surgery, and not just the result of the repair methods used, like stenting, dilatation, or surgery [83,84]. It is important to recognize that high blood pressure above the coarctation is the most probable cause of these vascular changes. This hypertension affects the arterial wall in two phases: initially, it causes the smooth muscle cells to hypertrophy, and later, there is an increase in fibrous elements in the wall [93].

Stenting has become a widely accepted treatment for aortic coarctation over the past two decades [94]. However, there is limited information on how the arteries change after this procedure. This lack of knowledge about the vascular changes following stenting underscores the need for continued research. Understanding the long-term effects of stenting is crucial for improving how patients are managed clinically.

Some studies suggest that stenting may lead to better blood flow and fluid dynamics than either surgical treatment or balloon dilation, especially when compared to healthy individuals [95]. Although stenting effectively eases the obstruction, improving overall function and reducing the size of the left ventricle, it is important to note that abnormalities in the structure and function of the arteries can still be present one year after the procedure [95].

In summary, a deeper understanding of blood flow dynamics and aortic stiffness across different treatment groups for aortic coarctation could enhance therapy. This knowledge would help develop better follow-up methods and refine surgical and interventional choices for these patients.

It is essential to note that adults undergoing coarctation repair tend to have a lower long-term survival rate than a matched normal population, with a notable decline starting in their thirties [80]. This significant observation emphasizes the need for tailored risk assessments to create personalized follow-up plans. By identifying each patient’s specific risk factors, follow-up routines can be customized. This approach aims to detect complications early and improve overall long-term health outcomes.

## 6. Recoarctation: Definitions and Thresholds for Reintervention

### 6.1. Contemporany Definition of Recoarctation

Based on the previous studies, recoarctation should be defined based on anatomical aspects, hemodynamic criteria, and clinical implications.

Anatomical Aspect: Recoarctation is characterized by a structural re-narrowing of the aorta at or adjacent to the site of initial surgical or catheter-based intervention for CoA. This anatomical re-narrowing is identifiable through imaging techniques such as echocardiography, magnetic resonance imaging (MRI), or computed tomography (CT) scans.

Hemodynamic Criteria: Hemodynamically, recoarctation is defined by a significant pressure gradient across the previously repaired coarctation site. A more than 20 to 30 mm Hg residual or recurrent pressure gradient threshold indicates significant recoarctation. This gradient is measurable non-invasively via Doppler echocardiography or invasively using cardiac catheterization.

Clinical Implications: Clinically, recoarctation may present with symptoms such as hypertension, claudication, or signs of heart failure. However, it can also be asymptomatic, detected only through routine follow-up or incidental imaging studies. The 2020 ESC guidelines for the management of ACHD stipulate that an increased pressure gradient (systolic ≥ 20 mmHg) between upper and lower extremities indicates recoarctation and warrants invasive assessment for confirmation and treatment [88].

### 6.2. Thresolds for Reintervention

A key indicator for reintervention is a peak-to-peak gradient of ≥20 mmHg across the coarctation site, as confirmed through invasive measurement. Reintervention may be considered in the presence of significant anatomical re-narrowing, specifically when there is ≥50% narrowing of the aorta relative to its diameter at the diaphragm. These thresholds apply irrespective of the patient’s blood pressure status, whether hypertensive or normotensive.

## 7. Our Experience-Case Series

In this case series, we present a collection of clinical cases showcasing our experience in the management of CoA in both pediatric and adult patients. These cases highlight various aspects of CoA, including initial diagnoses, recoarctation, and successful treatment interventions.

Case 1. Infant Coarctation

A 4-month-old infant underwent coarctation correction surgery using an extended end-to-end anastomosis. However, at 4 months, the infant developed hypertension and a blood pressure gradient of 40 mmHg, indicating recoarctation. This was precisely measured at 2 mm by a CT scan and confirmed by angiography. Post-treatment angiography after balloon dilatation showed a significant reduction in the gradient to 16 mmHg. Figure 2 illustrates the diagnostic process and treatment approach for this case.

Case 2. Adult-Onset Native Coarctation

A 23-year-old woman was incidentally diagnosed with native coarctation during a routine check-up for hypertension during pregnancy. The angioscanner revealed a 4 mm isthmic coarctation; aortic angiography confirmed a trans-coarctation gradient of 34 mmHg. She was successfully treated with a stent, resulting in optimal positioning in the aorta with no evidence of aortic wall damage. Figure 3 illustrates the diagnosis and management of this case.

Case 3. Pediatric Isthmic Recoarctation

A 6-month-old infant weighing 6.7 kg. experienced immediate recoarctation due to a clot on the anastomosis after undergoing coarctation repair at 8 days of life. A notch on the dilatation balloon indicated a trans-recoarctation gradient of 20 mmHg. After treatment with a 7 × 20 mm Tyshak balloon, the residual gradient was reduced to 9 mmHg. Figure 4 shows the diagnostic and therapeutic steps taken in the case of a pediatric patient with isthmic recoarctation.

Case 4. Infant Recoarctation after Initial Repair

Another case of a 4-month-old infant weighing 4.8 kg who had previously undergone treatment for tight isthmic coarctation using the Waldhausen procedure at 14 days of life following cardiogenic shock developed recoarctation. The recoarctation was measured at 3.7 mm, and post-treatment angiography showed a successful outcome with no residual gradient. Figure 5 presents the diagnostic and treatment process of this patient.

Case 5. Toddler’s Recoarctation after Initial Repair

A 3-year-old patient weighing 12 kg who had previously undergone a Crafoord extended procedure at 16 days of life, also experienced recoarctation. The recoarctation was successfully treated, resulting in a satisfactory post-dilatation outcome. Figure 6 outlines the diagnostic and therapeutic approach for this case.

## 8. Conclusions

In conclusion, the choice of repair technique for coarctation of the aorta significantly influences the occurrence of recoarctation. Surgical repair, which involves resection of the narrowed segment and direct anastomosis, often provides durable results with a low likelihood of recoarctation. On the other hand, percutaneous techniques like balloon angioplasty can offer less invasive options for certain patients but may have a higher risk of recoarctation, especially in younger individuals. Decisions should be made based on the patient’s age, anatomy, and overall health, focusing on achieving long-term success while minimizing the potential for re-narrowing of the aorta. Regardless of the chosen technique, regular follow-up and close monitoring are essential to ensure the best possible outcomes for individuals with coarctation of the aorta.

## Figures and Tables

**Figure 1 jcm-12-07350-f001:**
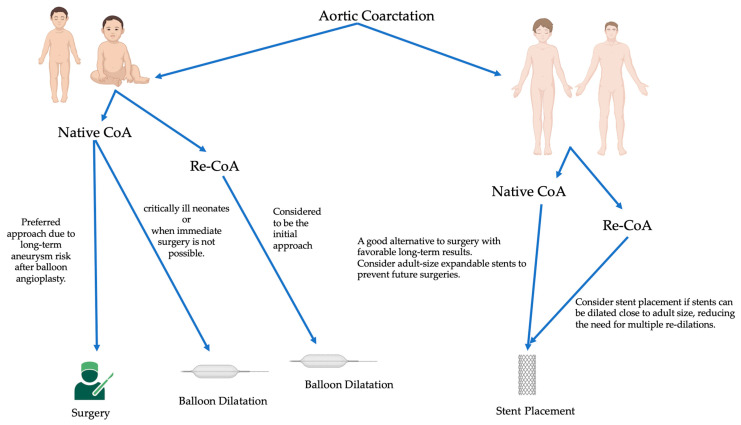
Management algorithm based on the age of the patient. Native CoA: native coarctation of the aorta; Re-CoA: recoarctation of the aorta.

**Figure 2 jcm-12-07350-f002:**
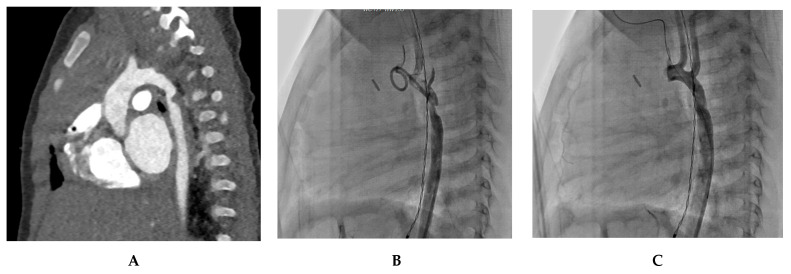
Infant coarctation case. (**A**) The CT scan highlights the recoarctation, precisely measuring it at 2 mm. (**B**) Angiography further confirms the recoarctation, showing a trans-coarctation gradient of 40 mmHg. (**C**) Post-treatment angiography after a 5.5 mm non-compliant Boston balloon dilatation, showing reduced gradient to 16 mmHg.

**Figure 3 jcm-12-07350-f003:**
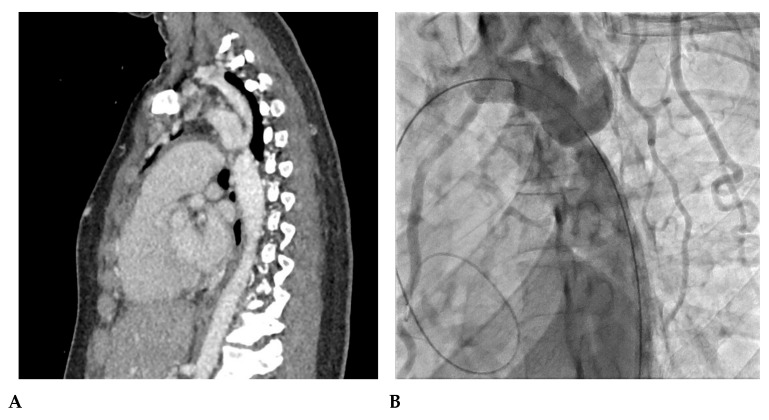
Adult-onset native coarctation. (**A**) The angioscanner reveals a 4 mm isthmic coarctation. (**B**) Confirmation through profile aortic angiography demonstrates a trans-coarctation gradient of 34 mmHg. (**C**) The patient was successfully treated with a 43 mm XXL Andrastent bare stent mounted on a 14 × 40 mm BiB Balloon, yielding ideal results. (**D**) The stent was initially placed and later redilated with a larger balloon to optimize its positioning in the aorta. The angiography clearly shows the bare stent properly as opposed to the aortic wall, with no evidence of aortic wall damage.

**Figure 4 jcm-12-07350-f004:**
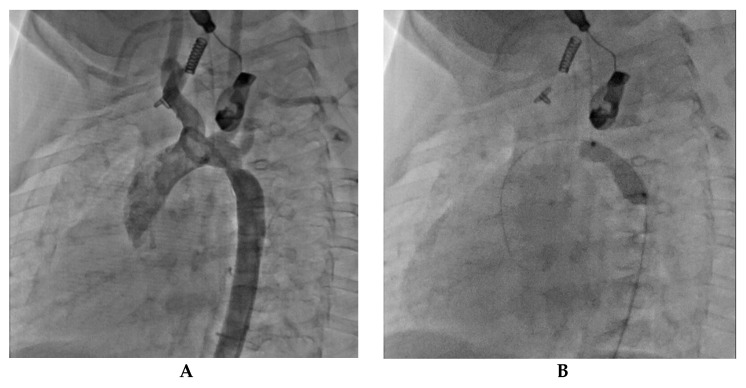
Pediatric isthmic recoarctation case. (**A**) Notch on the dilatation balloon. This notch reflects a trans-recoarctation gradient of 20 mmHg. (**B**) Close-up view of the notch on the dilatation balloon, highlighting the details of the recoarctation. (**C**) The notch was effectively and completely lifted by a 7 × 20 mm Tyshak balloon. (**D**) Subsequent angiography reveals good angiographic and hemodynamic results, with a residual gradient of just 9 mmHg.

**Figure 5 jcm-12-07350-f005:**
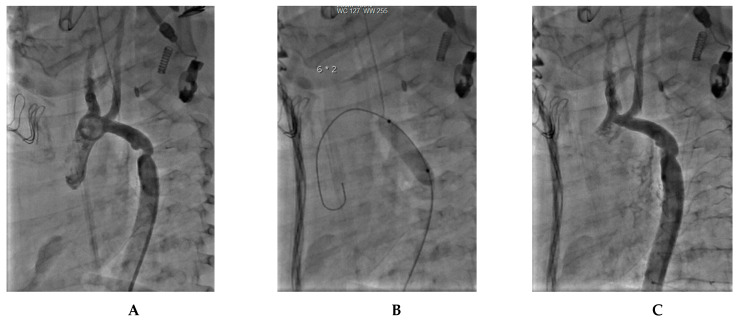
Infant recoarctation after initial repair (Waldhausen technique). (**A**) The recoarctation is measured at 3.7 mm, in the context of a horizontal aorta measuring 5.3 mm and a descending aorta of 7.1 mm. The trans-recoarctation gradient is 30 mmHg. Note the absence of a left subcardiac artery. (**B**) Treatment involved a 6 × 20 mm Tyshak balloon dilatation. (**C**) Subsequent angiography shows an isthmus diameter of 5.5 mm with no residual gradient, indicating a successful outcome.

**Figure 6 jcm-12-07350-f006:**
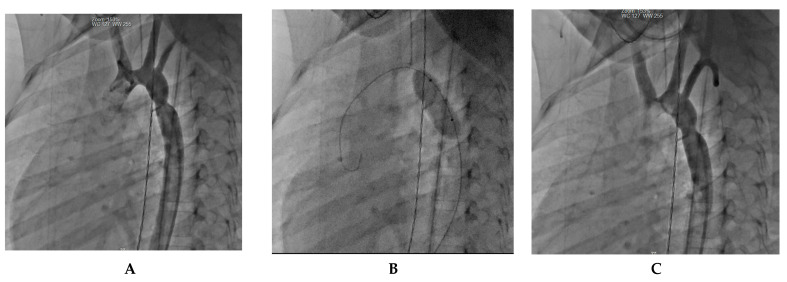
Toddler’s recoarctation after initial repair. (**A**) Isthmus before dilatation, measuring 5.4 mm within a descending aorta and aortic arch of 9.4 mm. (**B**) Dilatation was achieved using an 8 × 20 mm Tyshak balloon. (**C**) Post-dilatation control angiography indicates a successful outcome with an isthmus diameter of 7 mm.

**Table 1 jcm-12-07350-t001:** Summary findings of the studies on CoA.

Study	Patients	Follow-Up	Recurrence Rate	Primary Repair Methods	Key Findings and Recoarctation Repair Approach
Kapetein et al. (1994) [66]	109	30 years	5.8%	Classic, Extended end-to-end	Classic repair had higher long-term recoarctation rate. Extended repair (polypropylene) showed no recoarctation. Age < 6 months was a prognostic factor.
Dehaki et al. (2010) [67]	188	81.6 months	10%	Patch, End-to-end, Subclavian flap repair	Subclavian flap repair had the lowest recurrence rate. Age influenced recurrence rates, with a high rate in the 1–5-year age group.
Uguz et al. (2010) [68]	91	44 months	12.1%	Extended end-to-end, End-to-end	Neonates had a higher recoarctation rate. Infants had a lower restenosis rate.
Jahangiri et al. (2000) [69]	185	6.2 years	6%	Subclavian flap	Subclavian flap angioplasty showed excellent long-term outcomes. Hypoplastic arch was not the primary site of recoarctation.
Burch et al. (2009) [70]	167	-	-	End-to-end	Suture type influenced outcomes in neonates and infants.
Sen et al. (2018) [71]	75	-	-	Balloon, Surgery	Balloon coarctoplasty had a higher reintervention rate. Age influenced the choice of surgical technique.
Rao et al. (1995) [72]	29	4.5 years	-	Surgical, Balloon	Both surgical repair and balloon angioplasty had recoarctation cases.
Mohan et al. (2009) [73]	60	Short-term	-	Stent implantation	Stents effectively increased CoA diameter and reduced gradients in children. Weight did not significantly impact results.
Corno et al. (2001) [74]	141	30 years	Varies	Various surgical	Recurrence rates varied by surgical approach, with no recoarctation in adults.
Padalino et al. (2019) [75]	341	10.2 years	4.5%	Extended end to end anastomosis, patch, or conduit interposition	Low recurrence rate, and recurrences managed with percutaneous procedures.
Kaushal et al. (2009) [76]	201	5.0 years	4%	Extended end-to-end anastomosis	stent implantation was successful.
Presbitero et al. (1987) [77]	226	20 years	-	Various surgical	Late-onset hypertension was noted, and recurrence varied by the surgical approach.
Josep Rodes-Cabau et al. (2007) [78]	80	3 years		Surgical Repair (77% End-to-End Anastomosis)	Surgical patients had higher complications and longer hospitalization.

## Data Availability

Data are contained within the article.

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
