# Peer review of "From Crafoord’s End-to-End Anastomosis Approach to Percutaneous Interventions: Coarctation of the Aorta Management Strategies and Reinterventions"

_jcm, 2023, doi:10.3390/jcm12237350_

Round 1

Reviewer 1 Report

Comments and Suggestions for Authors

This is a comprehensive literature review of the treatment of aortic coarctation. 

the manuscript would be improved if the author's provided illustrations of the surgical repairs.

Part 4 of discussion seems laborious and redundant. It does not add to the summary that was already presented. 

Author Response

Dear Reviewer,

Thank you for your insightful comments and suggestions regarding our manuscript. We appreciate your valuable feedback and agree with your recommendations for enhancing the comprehensiveness and clarity of our paper.

This is a comprehensive literature review of the treatment of aortic coarctation. 

the manuscript would be improved if the author's provided illustrations of the surgical repairs.

We appreciate your suggestion, but unfortunately, our graphician was not able to illustrate with the 10 days we were allowed for the revision.  

Part 4 of discussion seems laborious and redundant. It does not add to the summary that was already presented. 

Our manuscript has been revised to eliminate de redundancy.

We have applied your suggestions seriously and our modified manuscript now presents a stronger and more convincing argument. Your detailed expertise has been essential in making this improvement.

Kind regards,

Corina Vasile

MD,PhD

Reviewer 2 Report

Comments and Suggestions for Authors

Vasile and colleagues present a broad review of coarctation interventions with focus on the treatment approach and outcomes for intervention on coarctation.  This review highlighted the differences between surgical and catheter based interventions for coarctation for patients from neonatal age through early adulthood with a discrete focus on recoarctation.

Major comments:

1.       Certainly, the theme or thrust of this review from this cardiology group was the adequacy of catheter based interventions compared to surgical interventions. The paper distinctly focused on simple coarctation treatment and the outcomes that were reviewed and reported across the literature only highlighted the recoarctation outcomes as the benchmark for success.  Although this is easy to tabulate, it oversimplifies the disease and limits the analysis to that which can be easily measured.

The longer term outcomes that really matter in coarctation are the restoration of adequate caliber of the aorta throughout its entire course and avoiding systemic implications like hypertension, early stroke, early MI.  With no mention of this bigger and more important aspect of coarctation including residual arch hypoplasia and systemic implications, the review comes across not as comprehensive but rather trying to make a point that catheter based interventions are equivalent to surgical based interventions. 

At the very least, it would improve the review to mention the implications of treatment strategy on the impact of outcomes of hypertension.  There are numerous references to cite in this regard.  This should include the mention of the implications of residual arch hypoplasia as an important decision maker on the treatment strategy as it has essential impact on the long term hypoplasia as well as stroke and MI outcomes.  These are three references to consider to that regard:

J Am Heart Assoc. 2015 Jun 25;4(7):e001978. doi: 10.1161/JAHA.115.001978.

Am J Cardiol. 2018 Dec 15;122(12):2120-2124. doi: 10.1016/j.amjcard.2018.08.051. 

J Am Heart Assoc. 2018 Jun 1;7(11):e009072. doi: 10.1161/JAHA.118.009072.

2.       The essential unanswered question with stent based treatment of coarctation are the implications on the complex impedance of the aorta with a stiff prosthesis in the aorta.  Although the diameter may be restored to near normal in these patient with primary intervention with stents, the resulting impedance of the aorta is no where near normal.  This also has long term implications for the afterload experienced by the ventricle and resulting secondary implications such as LV hypertrophy, etc.  It would be important to note this important potential long term implication and the research and opportunity to understand how it impacts treatment algorithms.3. The other unaddressed important clinical issue in long term assessment of coarctation patients is the definition of recoarctation and indications for treatment of recoarctation. Although many of the citations do not clearly address this, it is a centrally important topic for the field and one that is germane to a review almost solely focused on implications of treatment measured by recoarctation interventions. There is an opportunity to outline contemporary definitions for recoarctation and recommended thresholds for reintervention.

Minor comments:

Line 74 – needs a period after anomaly

Line 413 – needs a space before importantly

Author Response

Dear Reviewer,

Thank you for your insightful comments and suggestions regarding our manuscript. We appreciate your valuable feedback and agree with your recommendations for enhancing the comprehensiveness and clarity of our paper.

Vasile and colleagues present a broad review of coarctation interventions with focus on the treatment approach and outcomes for intervention on coarctation.  This review highlighted the differences between surgical and catheter based interventions for coarctation for patients from neonatal age through early adulthood with a discrete focus on recoarctation.

Major comments:

  1. Certainly, the theme or thrust of this review from this cardiology group was the adequacy of catheter based interventions compared to surgical interventions. The paper distinctly focused on simple coarctation treatment and the outcomes that were reviewed and reported across the literature only highlighted the recoarctation outcomes as the benchmark for success.  Although this is easy to tabulate, it oversimplifies the disease and limits the analysis to that which can be easily measured.

The longer term outcomes that really matter in coarctation are the restoration of adequate caliber of the aorta throughout its entire course and avoiding systemic implications like hypertension, early stroke, early MI.  With no mention of this bigger and more important aspect of coarctation including residual arch hypoplasia and systemic implications, the review comes across not as comprehensive but rather trying to make a point that catheter based interventions are equivalent to surgical based interventions.  

At the very least, it would improve the review to mention the implications of treatment strategy on the impact of outcomes of hypertension.  There are numerous references to cite in this regard.  This should include the mention of the implications of residual arch hypoplasia as an important decision maker on the treatment strategy as it has essential impact on the long term hypoplasia as well as stroke and MI outcomes.  These are three references to consider to that regard:

J Am Heart Assoc. 2015 Jun 25;4(7):e001978. doi: 10.1161/JAHA.115.001978.

Am J Cardiol. 2018 Dec 15;122(12):2120-2124. doi: 10.1016/j.amjcard.2018.08.051. 

J Am Heart Assoc. 2018 Jun 1;7(11):e009072. doi: 10.1161/JAHA.118.009072.

We have addressed your request.

  1. The essential unanswered question with stent based treatment of coarctation are the implications on the complex impedance of the aorta with a stiff prosthesis in the aorta.  Although the diameter may be restored to near normal in these patient with primary intervention with stents, the resulting impedance of the aorta is no where near normal.  This also has long term implications for the afterload experienced by the ventricle and resulting secondary implications such as LV hypertrophy, etc.  It would be important to note this important potential long term implication and the research and opportunity to understand how it impacts treatment algorithms.

We have added a paragraph that addresses this issue.

  1. The other unaddressed important clinical issue in long term assessment of coarctation patients is the definition of recoarctation and indications for treatment of recoarctation. Although many of the citations do not clearly address this, it is a centrally important topic for the field and one that is germane to a review almost solely focused on implications of treatment measured by recoarctation interventions. There is an opportunity to outline contemporary definitions for recoarctation and recommended thresholds for reintervention.

We have added a paragraph that addresses this issue.

Minor comments:

 Line 74 – needs a period after anomaly

We have revised it accordingly.

Line 413 – needs a space before importantly

We have revised it accordingly.

We have applied your suggestions seriously, and our modified manuscript now presents a stronger and more convincing argument. Your detailed expertise has been essential in making this improvement.

Kind regards,

Corina Vasile

MD,PhD

Reviewer 3 Report

Comments and Suggestions for Authors

The Authors have very well reviewed the management strategies and reinterventions of Coarctation of Aorta. Some changes needed:

1]  3.1.4 numbering has been used for Patch Angioplasty and Extended end-to-end anastomosis. Please rectify

2] Please add a paragraph on single vs stage-management approach for Coarctation of aorta when associated with other congenital cardiac anomalies.

Author Response

Dear Reviewer,

Thank you for your insightful comments and suggestions regarding our manuscript. We appreciate your valuable feedback and agree with your recommendations for enhancing the comprehensiveness and clarity of our paper.

The Authors have very well reviewed the management strategies and reinterventions of Coarctation of Aorta. Some changes needed:

1]  3.1.4 numbering has been used for Patch Angioplasty and Extended end-to-end anastomosis. Please rectify

Thank you for pointing this out. We have revised the numbering of the subtitles.

2] Write a paragraph on single vs stage-management approach for Coarctation of aorta when associated with other congenital cardiac anomalies.

We appreciate your suggestion. We have added a paragraph on this topic.

We have applied your suggestions seriously and our modified manuscript now presents a stronger and more convincing argument. Your detailed expertise has been essential in making this improvement.

Kind regards,

Corina Vasile

MD, PhD

Round 2

Reviewer 2 Report

Comments and Suggestions for Authors

Good additions to the paper related to my initial comments. 

As a surgeon, I don't agree with the added paragraphs about staging but those must have been added in response to other reviewer comments. Staging coarctation repair associated with other abnormalities results in increased procedures without increased benefit in many situations.  The arch reconstruction is the same and typically better when done with other procedures rather than done as an isolated procedure.  One would never do coarct, PA band, and then complete repair as three separate steps as was suggested in one sentence in the added paragraphs. It is a complex topic not well described by cardiologists in this review.  I personally would remove those paragraphs as they take away from the paper but I'll leave that up to the editor.  

Author Response

Dear reviewer,

Thank you once again for your valuable and constructive feedback.

Indeed, that paragraph has been added based on a reviewer's suggestion. We have revised the sentence you pointed out regarding the three separate steps.

At the editor’s demand, we could remove this paragraph if needed.

Kind regards,

Corina Vasile

MD,PhD